# Analysis of the Circulating Metabolome of Patients with Cutaneous, Mucosal and Uveal Melanoma Reveals Distinct Metabolic Profiles with Implications for Response to Immunotherapy

**DOI:** 10.3390/cancers15143708

**Published:** 2023-07-21

**Authors:** Maysa Vilbert, Erica C. Koch, April A. N. Rose, Rob C. Laister, Diana Gray, Valentin Sotov, Susanne Penny, Anna Spreafico, Devanand M. Pinto, Marcus O. Butler, Samuel D. Saibil

**Affiliations:** 1Princess Margaret Cancer Centre, University Health Network, Toronto, ON M5G 2M9, Canada; maysavilbert@gmail.com (M.V.); erica.kochhein@uhn.ca (E.C.K.); rob.laister@uhnresearch.ca (R.C.L.); diana.gray@uhnresearch.ca (D.G.); valentin.sotov@uhnresearch.ca (V.S.); anna.spreafico@uhn.ca (A.S.); marcus.butler@uhn.ca (M.O.B.); 2Department of Medicine, Division of Medical Oncology, University of Toronto, Toronto, ON M5S 1A8, Canada; 3Department of Hematology and Oncology, School of Medicine, Pontificia Universidad Católica de Chile, Santiago 8331150, Chile; 4Department of Oncology, Jewish General Hospital, Lady Davis Institute, McGill University, Montréal, QC H3G 2M1, Canada; april.rose@mcgill.ca; 5National Research Council, Human Health Therapeutics, Halifax, NS B3H 3Y8, Canada; susanne.penny@nrc-cnrc.gc.ca (S.P.); dev.pinto@nrc-cnrc.gc.ca (D.M.P.)

**Keywords:** melanoma, cutaneous melanoma, mucosal melanoma, uveal melanoma, immune-checkpoint inhibitors, metabolomics, tryptophan, kynurenine, spermine

## Abstract

**Simple Summary:**

Advanced melanoma is an aggressive cancer with a historical 12-month median survival after the diagnosis. This scenario has greatly changed since the advent of immunotherapy. Unfortunately, half of patients do not respond to this treatment and rare types of melanomas such as mucosal and uveal are known for their poorer outcomes compared to cutaneous melanoma. We investigated the serum metabolome of patients with these three melanomas subtypes, aiming to identify metabolic profiles that correlate with response to immunotherapy. Our results indicate that mucosal and uveal melanomas have a very distinct profile of circulating metabolites compared to cutaneous melanomas, mainly in the kynurenine pathway. Uveal melanoma, the most resistant subtype, also exhibits higher levels of sphingolipids and spermine. The later was associated with a poor survival in patients treated with immunotherapy. These results contribute to the growing evidence about tumoral resistance mechanisms to immunotherapy, with potential implication for future targeted therapies.

**Abstract:**

Cutaneous melanoma (CM) patients respond better to immune checkpoint inhibitors (ICI) than mucosal and uveal melanoma patients (MM/UM). Aiming to explore these differences and understand the distinct response to ICI, we evaluated the serum metabolome of advanced CM, MM, and UM patients. Levels of 115 metabolites were analyzed in samples collected before ICI, using a targeted metabolomics platform. In our analysis, molecules involved in the tryptophan-kynurenine axis distinguished UM/MM from CM. UM/MM patients had higher levels of 3-hydroxykynurenine (3-HKyn), whilst patients with CM were found to have higher levels of kynurenic acid (KA). The KA/3-HKyn ratio was significantly higher in CM versus the other subtypes. UM, the most ICI-resistant subtype, was also associated with higher levels of sphingomyelin-d18:1/22:1 and the polyamine spermine (SPM). Overall survival was prolonged in a cohort of CM patients with lower SPM levels, suggesting there are also conserved metabolic factors promoting ICI resistance across melanoma subtypes. Our study revealed a distinct metabolomic profile between the most resistant melanoma subtypes, UM and MM, compared to CM. Alterations within the kynurenine pathway, polyamine metabolism, and sphingolipid metabolic pathway may contribute to the poor response to ICI. Understanding the different metabolomic profiles introduces opportunities for novel therapies with potential synergic activity to ICI, to improve responses of UM/MM.

## 1. Introduction

Immune checkpoint inhibitors (ICI) have revolutionized the treatment of cancer, particularly for patients with cutaneous melanoma (CM). Treatment with the ICI combination of an anti-Programmed Death 1 (PD1) antibody and an anti-Cytotoxic T cell Late Antigen 4 (CTLA4) antibody has demonstrated a median overall survival (OS) of 72 months and an objective response rate greater than 50% in clinical trials in patients with advanced CM [1,2,3,4]. Unfortunately, for the rarer subtypes of melanoma, the impact of ICI has been more modest. These rarer subtypes of melanoma include those that arise from mucosal surfaces (mucosal melanoma, MM) as well as uveal melanomas (UM), which arise predominantly from the melanocytes which reside in the iris, ciliary body, and choroid of the eye. More limited data are available on the response to treatment with ICI therapy in patients with MM compared to CM [5,6,7]. Post-hoc analysis of multiple trials, however, has demonstrated that mucosal melanoma responds less well to treatment with both single-agent and combination ICI therapy as compared with cutaneous melanoma. For example, a combined analysis of six studies revealed that the objective response rate was 23% in patients with MM compared to 41% in patients with CM treated with ICI regimens [6]. For those treated with combination of anti-PD1 and anti-CTLA4, the progression-free survival (PFS) was 5.9 months in patients with MM and 11.7 months for CM patients, and the objective response rate was 37% versus 60%, respectively [6]. A similar lack of efficacy of ICI therapy has been observed in patients with UM. Two phase II studies suggested very limited activity of anti-PD1 therapy combined with anti-CTLA4 therapy in patients with UM, as objective responses were seen in only 18% of patients and median PFS and OS were 6 and 19 months [5,6]. Clearly, further investigation is warranted to interrogate the underlying mechanisms that account for the relative resistance to ICI therapy in patients with MM and UM as compared to patients with CM.

Emerging data has demonstrated that the circulating levels of various metabolites in the blood impact immune cell function and response to ICI therapy. In preclinical models, it was found that blood levels of the gut microbiome-derived metabolite inosine directly affected T cell activation and response to ICI therapy [8]. Similarly, serum levels of the metabolite trimethylamine N-oxide (TMAO) impacted macrophage phenotype, T cell activation, and response to ICI therapy in a murine pancreatic tumor model [9]. Moreover, in patients with advanced melanoma, there is accumulating data correlating the composition of the circulating metabolome with response to ICI. Higher pre-treatment levels of vitamin B5 (pantothenate) in the serum of patients with stage III and IV melanoma treated with anti-PD1 therapy were correlated with prolonged times to next treatment [10]. Dynamic changes in circulating ratios of tryptophan and its metabolites kynurenine were also demonstrated to correlate with survival in melanoma patients treated with PD1 blockade [11]. Additionally, a recent study employed untargeted metabolomic analysis of the plasma of CM patients before treatment with anti-PD1 therapy and defined metabolic profiles associated with response to treatment [12]. Finally, multiple studies have explored the interaction between the microbiome, metabolomics, and the immune system [13,14]. Consequently, metabolites have emerged as potential biomarkers for treatment response, particularly in relation to immune checkpoint inhibitors (ICI).

Given the divergent rates of response to ICI therapy, we hypothesized that the different subtypes of melanoma are associated with distinct alterations of the circulating metabolome, which impact the success of ICI therapy. Using a targeted metabolomics approach, we aimed to interrogate the serum metabolic profile from cutaneous, mucosal, and uveal melanomas patients to better understand their distinct responses to ICI therapy.

## 2. Materials and Methods

This is an exploratory retrospective observational cohort study of patients with advanced melanoma treated with ICI at Princess Margaret Hospital. Demographic and clinical data were collected from medical chart in a protected dataset. Clinical response was determined by investigator assessment of clinical and radiologic parameters. Patients were staged following the 8th edition of the American Joint Committee on Cancer staging system for cutaneous melanoma and uveal melanoma [15,16,17,18]. Serum samples were collected before immunotherapy with the approval of the University Health Network Research Ethics Board.

### 2.1. Metabolomics

For metabolic profiling of human serum, 20 μL of serum was added to 80 μL of HPLC grade methanol, and 5 μL of diluted ILIS standard (diluted 1:1 with HPLC grade methanol; Cambridge Isotope Laboratories Inc. product # MSK-A2-1.2, Tewksbury, MA, USA) was added to each sample which was then vortexed, incubated for 30 min at −80 °C to precipitate protein, and then centrifuged at 15,000× *g* to clarify. Then, 10 μL of supernatant was diluted in 990 μL of buffer containing 95% acetonitrile and 5% 20 mM ammonium carbonate (pH 9.8). Quality control samples (QCs) were prepared by pooling 10 μL of each sample. All samples, including QCs, were then analyzed by selected reaction monitoring (SRM) using a Waters XBridge Amide 2.1 × 50 mm, 3.5 μm particle size column using acetonitrile/ammonium carbonate (pH 9.8) acetonitrile buffer system coupled with a Sciex Qtrap 5500 triple quadrupole linear ion trap tandem mass spectrometer.

The data acquisition included 317 transitions. Data were captured using Analyst, version 1.6.2 software (Sciex); peak integration was performed using Skyline, version 4.1 [19]. An in-house R script was used for data QC analysis and normalization (Version 3.1.2, http://www.r-project.org, accessed on 15 July 2021).

### 2.2. Statistical Analysis

Statistical analysis was performed using the MetaboAnalystR package in R software Version 3.1.2 [20]. Differences in metabolomics profiles were compared using Random Forest (RF), Multidimensional scaling (MDS), and U-Mann-Whitney test. We used analysis of variance (ANOVA) to determine the differences in hydroxykynurenine between cutaneous, mucosal, and uveal melanomas. The Kruskal–Wallis test was performed on the kynurenic acid, kynurenine/tryptophan ratio, and kynurenic acid/hydroxykynurenine ratio. Overall survival (OS) was assessed using Kaplan-Meier, Log-Rank, and Cox regression models. Statistical significance was set to 0.05. All statistical tests were two-sided. Statistics and relevant information as *p*-values are reported in the figures and associated legends. Statistical analyses were done using IBM SPSS Statistics, Version 28.0.0.0.

## 3. Results

### 3.1. Serum Metabolomic Profiling Reveals Distinctions between Melanoma Subtypes

To interrogate the differences in the circulating metabolome between the different subtypes of melanoma, we obtained pre-ICI serum samples from 36 patients: 13 with CM, 12 with MM, and 11 with UM. All the participants had stage IV disease and received subsequent treatment with single agent anti-PD1 or anti-CTLA4. One participant underwent resection of the metastatic site and received adjuvant immunotherapy, in the MM group, whilst the rest of the patients received ICI therapy in the metastatic setting. The population demographics, including biological sex, age, mutational status as well as clinical and radiological response to ICI of the cohort are described in Table 1. Of note, none of the patients received treatment with combined PD-1 and CTLA-4 blockade. One patient with MM transferred care to another center and was not included in the clinical response data.

These pre-ICI serum samples were then assayed for the relative abundance of 115 metabolites using a targeted mass spectrometry approach. Analysis of this targeted metabolomics data using unsupervised hierarchical clustering (Figure 1a) or multidimensional scaling (MDS) by Random Forrest (RF) analysis (Figure 1b) demonstrated the three melanoma subtypes clustered according to metabolic profile, albeit more distinctly using RF analysis. The relative discriminatory potential of specific metabolites selected by the RF model is provided in Figure 1c. Many of the metabolites whose abundance distinguished the circulating metabolome of patients with CM, MM, and UM were lipids, including multiple different choline-containing phospholipids as well as species of sphingolipids. The serum levels of many of these lipids were lower in patients with MM and higher in patients with CM and UM, particularly in the context of the various phosphatidylcholine-containing lipids (Figure 1c and Appendix A). These data suggest that lipid metabolism, and in particular phosphatidylcholine metabolism, may be differentially regulated between MM and the other melanoma subtypes. Importantly, however, these data also demonstrate clear differences in the circulating metabolome amongst patients with different subtypes of melanoma and supports our hypothesis that these differences could contribute to the variable responses to ICI therapy amongst the different types of melanomas. 

### 3.2. Differences in the Serum Levels of Metabolites Involved in the Tryptophan-Kyneurine Pathway Are Observed between Cutaneous versus Mucosal and Uveal Melanomas

Next, we attempted to identify the metabolites whose levels might be associated with response to ICI therapy. To do this, we first grouped UM and MM together (UM/MM) due to their relative resistance to ICI therapy and compared the serum metabolite profiles of this combined group with the CM patients. Using hierarchical clustering and RF analysis, we noted that molecules involved in the tryptophan–kynurenine metabolic axis distinguished the combined UM/MM group from CM. We observed differences in the levels of kynurenic acid (KA) and 3-hydroxykynurenine (3-HKyn) between the two groups (Figure 2a,b). The kynurenine pathway (KP) involves the conversion of tryptophan (Tryp) to kynurenine (Kyn) and then ultimately to nicotinamide adenine dinucleotide (NAD+) via a multi-step pathway involving multiple intermediary metabolites (Figure 2c). The first step of the KP involves the conversion of tryptophan into N-formylkynurenine by the enzymes indoleamine 2,3-dioxygenase 1 (IDO1), indoleamine 2,3-dioxygenase 2 (IDO2), or tryptophan 2,3-dioxygenase (TDO). N-formylkynurenine is then rapidly converted into L-kynurenine which then can be further metabolized via one of three branches. L-Kynurenine can be metabolized into (3-HKyn) via the activity of kyneurine 3-monooxygenase (KMO) down the major branch of the pathway. Alternatively, l-kynurenine can be converted into KA via the activity of kyneurine aminotransferases (KAT) enzymes or into anthranilic acid (AA) via the activity of kyneurinase (KYNU) [21]. Activation of the KP is associated with suppression of the anti-tumor T cell response via local depletion tryptophan availability in the tumor microenvironment (TME) by IDO1, IDO2, or TDO as well as the generation of suppressive myeloid and T regulatory cells via ligation of the aryl hydrocarbon receptor (AHR) by Kyn and its metabolites [22,23]. Recently, however, additional immunomodulatory effects of KP metabolites, such as 3-HKyn, independent of activating AHR have been described indicating a more complex role of the KP in regulating the immune response [21,24]. Moreover, in a cohort of patients with non-small cell lung cancer (NSCLC), it has been reported that increased levels of three downstream KP metabolites, 3-HKyn, anthranilic acid, and quinolinic acid, are correlated with inferior responses to ICI therapy [25]. Accordingly, this prompted us to interrogate the activity of different steps in the KP in more detail in patients with each subtype of melanoma.

Previously, the ratio of the levels of Kyn to Tryp (Kyn/Tryp) in the circulation was used as a measure of the relative activity of the first steps of the KP pathway [11]. In our cohort, the mean values for Kyn/Tryp ratio were higher in patients with CM than in patients with MM or UM, although only the difference between CM and UM patients was statistically significant (Figure 2d). The circulating levels of Tryp and Kyn, however, were not significantly different amongst patients with the three subtypes of melanoma (Figure 2d). Conversely, significantly higher levels of 3-HKyn were found in the samples from patients with UM and MM as compared to CM, whilst patients with CM were found to have higher levels of serum KA compared the other subtypes (Figure 2e). Comparing the ratio of KA to 3-HKyn, it was 3.1 for CM (standard deviation 1.8), 1.2 for MM (standard deviation 0.42), and 1.3 for uveal melanoma (standard deviation 0.45) (Figure 2e). Collectively, these findings suggest an increase in KAT activity versus KMO activity in patients with CM vs MM and UM which results in increased amounts of KA versus 3-Hkyn in the circulation. These data also imply differential utilization of the branches of the KP may be a metabolic discriminator between CM versus MM and UM and contribute to the poor response to ICI of patients with MM and UM.

### 3.3. Sphingomyelin and Polyamine Metabolites Correlate with Resistance to ICI Therapy

Finally, to attempt to identify other metabolites that might associate with resistance to ICI therapy, we compared the UM to patients, the most ICI-resistant melanoma subtype, with the grouped CM and MM patients. Using unsupervised hierarchical clustering and RF analysis, we found that multiple metabolites that were more abundant in the serum of UM patients versus the other subtypes (Figure 3a,b). Amongst these metabolites that were higher in the circulation of the UM patients were two sphingolipids: sphingomyelin (d18:1/22:1(13Z) and sphingomyelin (d18:1/16:0) (Figure 3c). Consistent with this observation, it was recently reported that higher levels of sphingomyelin species in the plasma were associated with resistance to anti-PD1 therapy in a cohort of CM patients [12]. Collectively, these findings suggest that there may indeed be metabolic pathways associated with resistance to ICI that are common to all the subtypes of melanoma but enriched in UM given its inherent resistance to ICI. In line with this hypothesis, we observed higher circulating levels of the polyamine spermine (SPM) in the patients with UM (Figure 3d). This enrichment of SPM in the serum of patients with UM was of interest to us as the anti-tumor activity of T cells and the efficacy of anti-PD1 therapy has recently been reported to be augmented by supplementation of the SPM polyamine precursor, spermidine (SPD) in pre-clinical models [26]. This study also demonstrated that high SPM levels were able to block the ability of SPD to enhance T cell function and enhance the efficacy of PD-1 blockade. Therefore, we aimed to investigate if higher levels of SPM in the circulation correlated with worse outcomes in patients with CM treated with ICI.

We compared the overall survival (OS) in response to ICI therapy of the CM patients in our cohort who had high levels of SPM in the circulation (top 30%) versus the lower SPM levels. Within our small cohort, there was a trend toward decreased OS with higher SPM levels, as the patients with the high SPM levels had a median OS of 6.1 months (1.4 to 10.8) versus 12.8 months (95% CI 0.8 to 24.8) for the patients with lower baseline SPM levels (Appendix A). To investigate if this trend would be validated in a larger patient sample, we correlated the baseline SPM levels with OS in a cohort of 77 CM patients treated with anti-PD1 therapy whose clinical outcomes and baseline metabolomic profiles were published previously [11]. Within this Li et al. (2019) cohort, we found that the patients with elevated baseline levels of circulating SPM (top 30%) had reduced median OS compared to the patients with lower baseline levels of SPM (Figure 3e). This difference was statistically significant with a median OS of 11.1 months (95% CI 5.57 to NR) for the SPM high group versus 24.9 months (95% CI 21.1 to NR) for the low SPM group. These data indicate that high serum SPM levels, a distinguishing characteristic of patients with UM, also correlates with inferior clinical outcomes in patients with CM treated with ICI.

## 4. Discussion

It is becoming increasingly apparent that the circulating metabolome influences multiple facets of the immune system, including response to ICI therapy. Here, we investigated if three subtypes of melanoma with varying responsiveness to ICI therapy were associated with different systemic metabolomic profiles. We observed differences in the level of multiple serum metabolites which could indeed be used to segregate the three subtypes of melanoma studied, CM, MM, and UM (Figure 1). Amongst the metabolites whose levels differed the most between subtypes, we identified molecules from three different metabolic pathways that all have been previously associated with response to ICI therapy. These pathways included the KP as well as sphingomyelin and polyamine metabolism. We found that higher serum levels of certain sphingolipid species, such as sphingomyelin (d18:1/22:1(13Z) and sphingomyelin (d18:1/16:0), and the polyamine SPM to be enriched in the serum of patients with UM, the most-ICI resistant melanoma subtype. Furthermore, when we analyzed SPM levels in a separate cohort of CM patients treated with anti-PD1 therapy, elevated circulating SPM levels significantly correlated with poorer overall survival (Figure 3). These data suggest that although specific melanoma subtypes may differentially impact the circulating metabolome, the impact on immune cell function and ICI-response as a result of the changes in the circulating levels of at least some of these metabolites is likely independent of melanoma subtype. Accordingly, further studies are warranted to both better understand the impact of each of these metabolic pathways on the immune system and to further delineate the mechanisms through which each subtype of melanoma uniquely impacts the circulating metabolome.

Of the three metabolic pathways our findings identified, the KP is perhaps the best studied in the context of ICI-therapy for melanoma. It has been reported that increases in the serum kynurenine/tryptophan (Kyn/Tryp) ratio are associated with worse survival in a cohort of patients with CM treated with anti-PD1 therapy [11]. This finding was interpreted to indicate that the induction of IDO1 or TDO activity is associated with adaptive resistance to ICI therapy and suggested that inhibition of IDO1 activity could enhance the efficacy of anti-PD1 therapy. In a phase III trial of combined PD-1 blockade and an IDO1-specific inhibitor epacadostat, however, the addition of the IDO1 inhibitor did not demonstrate any clinical benefit [27]. Although the reasons for the lack of efficacy observed in the epocadostat trial remain controversial, the failure of this trial has prompted a re-evaluation of IDO1 as a therapeutic target [28]. Our data suggests that, at least for MM and UM, other enzymes in the KP may be attractive alternative therapeutic targets in the KP. We did not observe large differences in the baseline serum levels of Tryp or Kyn or in the ratio of Kyn/Tryp amongst the three melanoma subtypes. We did, however, observe increased levels of 3-HKyn and decreased levels of KA in the MM and UM patients versus the patients with CM (Figure 2). This resulted in a significantly decreased ratio of KA/3-HKyn in the patients with UM and MM and suggested that, in these patients, Kyn is predominantly being converted to 3-HKyn via the activity of the enzyme KMO versus being converted to KA via KATs. Thus, there may be differential regulation of the branches of the KP between patients with CM and those with MM and UM. This observation mirrors what has recently been reported for breast cancer, as it was suggested that the Her2+ and the triple negative subtypes of breast cancer are associated with increased expression of KMO and KNYU and utilization of those branches of the KP versus patients with luminal breast cancer [29]. Additionally, given our increasing understanding of the underlying potent immunosuppressive effects of 3-HKyn, our data also suggest that increased 3-HKyn levels could contribute to the decreased responsiveness of MM and UM to ICI therapy [21,24]. Accordingly, there might be a rationale for combining KMO-specific inhibitors with ICIs in the treatment of UM and MM [30].

Growing scientific evidence has demonstrated the relationship between the intestinal microbiome and the response to ICI [31]. Microbiome-derived metabolites seem to play an important role in modulating immunity. Several studies have explored and demonstrated the association of short-chain fatty acids (SCFAs), such as acetate, propionate, and butyrate, with PFS in patients treated with ICI [32,33,34]. Regarding the kynurenine pathway, the essential amino acid tryptophan, derived from the diet, is mainly absorbed in the small intestine; however, the proportion that reaches the colon is metabolized by gut-derived microbiota into metabolites [35]. This promising field can contribute to more personalized care, and further research focused on biomarkers to identify early progressors or long-responder patients is eagerly awaited.

In addition to metabolites of the KP, we also observed higher serum levels of sphingomyelin species in patients with uveal melanoma compared to the other subtypes (Figure 3). As mentioned above, high circulating levels of sphingomyelin species have recently been linked with resistance to ICI therapy in patients with CM [11]. The mechanisms through which cancers alters sphingolipid metabolism and how these alterations result in resistance to ICI therapy have not been fully characterized [36]. However, a major step in sphingolipid metabolism is the conversion of sphingolipid species to ceramide, a molecule that appears to suppress tumor growth rather than promote it [37]. In keeping with this, it has recently been found that, in patients with CM, lower expression of the enzyme that converts sphingolipids to ceramide, neutral sphingomyelinase 2 (nSMase2), was associated with worse survival [38]. Moreover, in a murine model, over-expression of nSMase2 in melanoma cells resulted in enhanced susceptibility to anti-PD1 therapy. Given these findings, future studies aimed at interrogating the expression of nSMase2 in a larger cohort of patients with UM are warranted. Additionally, the expression of nSMase2 should be correlated with circulating levels of sphingomyelin species, and response to ICI therapy in patients with UM as expression of this enzyme may link the failure of ICI therapy in patients with UM to serum sphingolipid levels.

Finally, the polyamine SPM was found to be elevated in the serum of patients with UM versus MM and CM. We also found that high serum levels of SPM were associated with decreased overall survival in a larger cohort of patients treated with ICI therapy. These data are in agreement with the growing literature detailing the effects of polyamine levels on the immune system and response to ICI therapy [39]. In addition to the recent study cited above [26] which directly links SPM and SPD with T cell metabolism and anti-tumor function, other preclinical studies have found that inhibitors of polyamine synthesis synergize with ICI therapy to increase tumor control [40]. It remains unclear, however, the extent to which modulating SPM levels is specifically responsible for the observed synergy with anti-PD1 therapy or if the response is driven mainly by an alteration in the levels of other molecules in the polyamine biosynthetic pathway. Unfortunately, we were unable to accurately measure the levels of SPD and other metabolites related to polyamines synthesis. As such, we cannot comment on the levels of these molecules in the serum of UM patients versus the other melanoma subtypes. Further studies utilizing a focused mass spectrometry methods optimized to interrogate the levels of molecules related to the polyamine synthesis pathway will be required to better address this question. Despite this uncertainty, our results are clearly consistent with the emerging literature that suggests targeting specific nodes in polyamine biosynthesis may serve to increase the response rates to anti-PD1 therapy in melanoma patients, regardless of subtype.

Our study has some limitations. First, we did not have direct evidence of the association between KP metabolites and resistance to ICI. This hypothesis was formed based on our observation of KP metabolites expression among different melanoma subtypes. Second, we could not identify significant differences in OS based on KP metabolites likely due to our small sample size. Therefore, further prospective and larger studies are needed to establish definitive conclusions.

## 5. Conclusions

Our analyses revealed significant differences in the metabolomic profile of uveal melanoma, mucosal melanoma, and cutaneous melanoma patients. Differential abundances of molecules involved in the Kyneurine pathway distinguished uveal and mucosal melanoma from cutaneous melanoma, whereas molecules involved in polyamines and sphingomyelin metabolism distinguished uveal melanoma from the other subtypes of melanoma. In addition, high spermine levels seems to be a prognostic factor of decreased response to ICI. Kynurenine pathway and polyamines are fundamental for intracellular energy production, cell replication, and modulation of the immune microenvironment, while the sphingolipid metabolism is important for cell differentiation and regulation of cell death (apoptosis) or proliferation (aggressiveness and invasion). The distinct metabolomic profiles of the three melanoma subtypes we studied allude to potential mechanisms underlying resistance to ICI therapy. This study also highlights the potential utility of profiling circulating metabolites in the discovery of novel targets and the development of combination therapies that may improve responses to ICI.

## Figures and Tables

**Figure 1 cancers-15-03708-f001:**
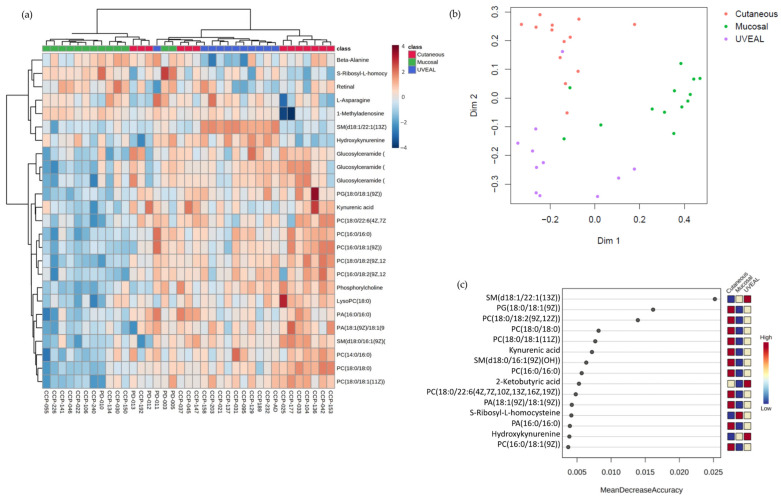
(**a**) HeatMap demonstrating the top 25 serum metabolites differentiating the melanoma subtypes by ANOVA analysis (**b**) MDS using RF analysis revealed distinct clustering of the three melanoma subtypes based upon serum metabolome. (**c**) Variable importance plot demonstrating the 15 most important metabolites to discriminate between the melanoma subtypes.

**Figure 2 cancers-15-03708-f002:**
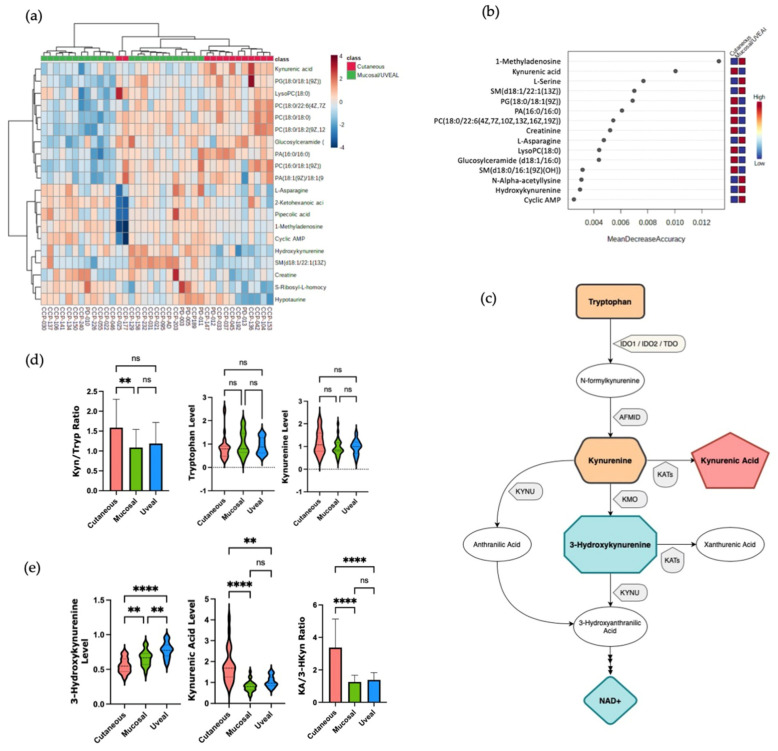
(**a**) HeatMap demonstrating the metabolites differences between cutaneous melanoma versus mucosal and uveal. (**b**) Random Forest analysis, cutaneous versus mucosal and uveal melanomas presented with distinct levels of 3-hydroxykynurenine and kynurenic acid. (**c**) Scheme of Kynurenine Pathway. (**d**) Conversion rate (mean, SD) estimated by kynurenine/tryptophan ratio (Kyn/Tryp), levels of tryptophan, and levels of kynurenine according to melanoma subtypes. (**e**) Levels of 3-hydroxykynurenine and kynurenic acid by melanoma subtypes, and kynurenic acid/3-hydroxykynurenine (KA/3-HKyn) ratio. ns = not statistically significant; ** *p* < 0.01; **** *p* < 0.0001.

**Figure 3 cancers-15-03708-f003:**
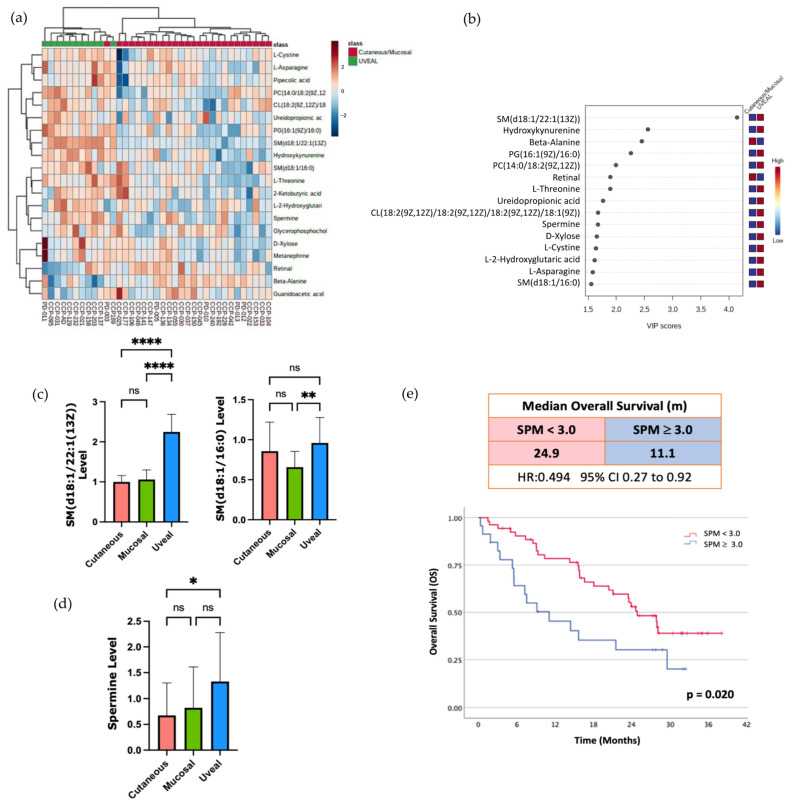
(**a**) HeatMap showing the metabolites differences between uveal melanoma versus cutaneous and mucosal. (**b**) Random Forest analysis, uveal melanoma versus cutaneous and mucosal melanomas demonstrated distinct levels of metabolites. (**c**) Sphingomyelins (SM) levels according to melanoma subtypes (mean, SD). (**d**) Spermine (SPM) levels in each of the three subtypes (mean, SD). (**e**) Overall survival analysis of melanoma patients treated with ICI, comparing low and high levels of spermine in a published external cohort of cutaneous melanoma patients [11]. Spermine cut-off point used for analysis was in the 70th percentile. ns = not statistically significant; * *p* < 0.05; ** *p* < 0.01; **** *p* < 0.0001.

**Table 1 cancers-15-03708-t001:** Characteristics of patients with cutaneous melanoma (CM), mucosal melanoma (MM), and uveal melanoma (UM).

Characteristics	CM (*n* = 13)	MM (*n* = 12)	UM (*n* = 11)
Age (years)—median (max–min)	57 (40–80)	62 (19–80)	56 (40–76)
Sex—*n* (%)			
Male	10 (76.9)	2 (16.7)	5 (45.5)
Female	3 (23.1)	10 (83.3)	6 (54.5)
M Stage—*n* (%)			
M1a	3 (23.1)	2 (16.7)	3 (27.3)
M1b	4 (30.8)	3 (25.0)	7 (63.6)
M1c	3 (23.1)	5 (41.7)	1 (9.1)
M1d	3 (23.1)	2 (16.7)	-
LDH (*n* = 25)			
Normal (ULN 220)	3 (42.9)	5 (62.5)	1 (10.0)
<1.5x	3 (42.9)	2 (25.0)	4 (40.0)
>1.5x	1 (14.3)	1 (12.5)	5 (50.0)
Tumor mutation—*n* (%)			
BRAF	3 (23.1)	2 (16.7)	-
KIT	-	1 (8.3)	-
NRAS	2 (15.4)	-	-
Wild type	8 (61.5)	9 (75.0)	11 (100)
Treatment—*n* (%)			
Anti-PD1	9 (69.2)	7 (58.3)	2 (18.2)
Anti-CTLA4	4 (30.8)	5 (41.7)	9 (81.8)
Line of therapy—*n* (%)			
Adjuvant	-	1 (8.3)	-
First	10 (76.9)	2 (16.7)	3 (27.3)
Second	2 (15.4)	6 (50.0)	6 (54.5)
Third	1 (7.7)	2 (16.7)	1 (9.1)
Fourth	-	1 (8.3)	1 (9.1)
Clinical Response—*n* (%)			
CR	1 (7.7)	1 (8.3)	-
PR	1 (7.7)	-	1 (9.1)
SD	4 (30.8)	-	3 (27.3)
PD	7 (53.8)	10 (83.3)	7 (63.3)

*n*, number; M, metastasis; ULN, upper limit of normality; CR, complete response; PR, partial response; SD, stable disease; PD, progressive disease.

## Data Availability

The data presented in this study are available on request from the corresponding author. The data are not publicly available due to protection of patient information.

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
