# Peer review of "Analysis of the Circulating Metabolome of Patients with Cutaneous, Mucosal and Uveal Melanoma Reveals Distinct Metabolic Profiles with Implications for Response to Immunotherapy"

_cancers, 2023, doi:10.3390/cancers15143708_

Round 1

Reviewer 1 Report

Vilbert at al. performed metabolomic profiling of serum samples from melanoma patients, and reported metabolomic differences in patients with various melanoma subtypes. Authors noted differential expression of components of tryptophan-kynurenine pathway and sphingolipid molecules in CM as compared to ICI resistant MM and UM. Following these results, authors observed better overall survival with ICI treatment in patients with low SPM levels. While these observations are reported in other studies in the past, the findings in the current report are mostly complementary to the current understanding in the field. Authors must address following comments.

1.    Although data is interesting, sample size is too small in this study to draw conclusion with confidence. Also, you do not have any direct evidence in this report to connect KP pathway to ICI resistance, except that positive correlation between KP pathway changes and MM/UM that are inherently resistant to ICI. Authors need to address this point.

2.    In line 199-200, authors mentioned that “These data suggested a slight increase in the conversion of Tryp to Kyn in CM versus MM and UM patients”. If tryptophan to Kyn conversion is high in CM, how would you explain lack of difference in Tryptophan levels in CM vs MM/UM. Alternatively, high Kyn levels could be established due to slow/delayed conversion of Kyn to 3-Hkyn in CM. Looks like Fig 2E supports this hypothesis. Have you looked at the expression of KMO/KAT in tumor cells?

3.    In figure S2 and 3E, what was the rationale for setting SPM cut off at 30% level?

Author Response

Dear Reviewer 1,

We are very appreciative of the careful reading which you gave our manuscript.  We meticulously addressed each of the points and questions raised and performed the corrections or justifications accordingly. In the discussion that follows, we have indicated our direct responses to the questions raised by your thorough review. In addition, we have highlighted in yellow the manuscript changes made according to your suggestions. We believe the manuscript is significantly improved because of your constructive criticisms and hope that you will find the revised version of this manuscript acceptable. 

Vilbert at al. performed metabolomic profiling of serum samples from melanoma patients, and reported metabolomic differences in patients with various melanoma subtypes. Authors noted differential expression of components of tryptophan-kynurenine pathway and sphingolipid molecules in CM as compared to ICI resistant MM and UM. Following these results, authors observed better overall survival with ICI treatment in patients with low SPM levels. While these observations are reported in other studies in the past, the findings in the current report are mostly complementary to the current understanding in the field. Authors must address following comments.

  1. Although data is interesting, sample size is too small in this study to draw conclusion with confidence. Also, you do not have any direct evidence in this report to connect KP pathway to ICI resistance, except that positive correlation between KP pathway changes and MM/UM that are inherently resistant to ICI. Authors need to address this point.
  2. In line 199-200, authors mentioned that “These data suggested a slight increase in the conversion of Tryp to Kyn in CM versus MM and UM patients”. If tryptophan to Kyn conversion is high in CM, how would you explain lack of difference in Tryptophan levels in CM vs MM/UM. Alternatively, high Kyn levels could be established due to slow/delayed conversion of Kyn to 3-Hkyn in CM. Looks like Fig 2E supports this hypothesis. Have you looked at the expression of KMO/KAT in tumor cells?
  3. In figure S2 and 3E, what was the rationale for setting SPM cut off at 30% level?

Thank you very much for your thoughtful review. Please, see below our answers to your concerns.

  1. We agree with the reviewer’s comment. We do not have direct evidence of the association of Kynurenine Pathway metabolites and resistance to ICI, and our sample size is small to draw definitive conclusions. Hence, in the final of the Discussion section, we added a paragraph informing the reader about these limitations in our study. Besides that, we have modified the text and references to include a recent study in lung cancer patients in which a similar correlation between blood levels of KP metabolites and resistance to ICI was observed to provide further rationale for our hypothesis.
  2. We appreciate this significant observation and agree that it may not be a case of high tryptophan-to-kynurenine conversion in cutaneous melanoma, but instead, a slow conversion of kynurenine-to-hydroxykynurenine. As such, we have removed this statement from the text as we do not have evidence to support which of these two possibilities explains our observations. Unfortunatelly, we did not look at the expression of KMO/KAT in tumor cells. 
  3. Given the nature of our exploratory analysis, we arbitrarily inferred the SPM cut-off level at 30% based on the data distribution after analyzing it by histogram distribution of the whole sample and by box plot distribution of uveal melanoma (group of interest) and of cutaneous plus mucosal melanoma (control group). Interesting that we established the cut-off point based on our small sample and observational plot analyses. Then, when the same rationale was applied to an external dataset with a bigger sample size, it held true, and we found significant and similar results to our study.

Again, we thank the reviewers for their suggestions and important contribution to improving our manuscript.

Sincerely,

Reviewer 2 Report

This study is an exploratory retrospective observational cohort study of patients with advanced melanoma treated with Immune checkpoint inhibitors. The authors investigated the serum metabolome of patients with mucosal, uveal or cutaneous melanoma, aiming to identify metabolic profiles that correlate with response to immunotherapy. They found different abundances of molecules involved in the Kyneurine pathway between uveal and mucosal melanoma from cutaneous melanoma and described that uveal melanoma, the most resistant melanoma subtype, exhibits higher levels of sphingolipids and spermine. Finally, high spermine levels seems to be a prognostic factor of decreased response to immunotherapy. 

The article is well written and the research very interesting and well described.

Author Response

Dear Reviewer 2,

We really appreciate your thoughtful review.

Thank you.

Reviewer 3 Report

Dear authors,

After the review process, I have several comments: statistical data should be inserted as a separate section; 

part of the figures (e.g., v) should be moved as supplementary files; in the discussion, the metabolomic pattern should be correlated with the microbiome pattern.

Best regards!

Author Response

Dear Reviewer 3,

We are very appreciative of the careful reading which you gave our manuscript.  We meticulously addressed each of the points and questions raised and performed the corrections or justifications accordingly. In the discussion that follows we have indicated our direct responses to the questions raised by your thorough review. In addition, we have highlighted in yellow the manuscript changes made according to your suggestions. We believe the manuscript is significantly improved because of your constructive criticisms and hope that you will find the revised version of this manuscript acceptable.

Dear authors,

After the review process, I have several comments: statistical data should be inserted as a separate section; Part of the figures (e.g., v) should be moved as supplementary files; in the discussion, the metabolomic pattern should be correlated with the microbiome pattern.

Best regards!

Thank you for your comments. We modified the methods inserting the statistical data as a separate section and added to the discussion about the correlation of metabolomic and microbiome patterns, as suggested. We also improved the introduction and references.

Again, we thank the reviewers for their suggestions and important contribution to improving our manuscript.

Sincerely,

Round 2

Reviewer 1 Report

Authors have addressed comments raised in the initial review. I suggest authors fix some minor language issues in the text. I do not have any further comments.

1.    Line 205-206: “Previously, the ratio of the levels of Kyn to Tryp (Kyn/Tryp) in the circulation has used….” I am not sure it’s right to say “…has used…”

2.    Line 213-214: “Comparing the ratio of KA to 3-HKyn it was 3.1…..” I would put “,” after “Comparing the ratio of KA to 3-HKyn”.

3.    Line 291-292: “Furthermore, when we analyzed SPM levels in a separate cohort of CM patients treated with anti-PD1 therapy elevated circulating SPM levels significantly correlated 292 with poorer overall survival (Figure 3)”. I would use I would put “,” after “Furthermore, when we analyzed SPM levels in a separate cohort of CM patients treated with anti-PD1 therapy”

4.    Line 323-324: “Additionally, given our increasing understanding un-323 derlying the potent immunosuppressive effects of 3-HKyn”. I am not sure this expression is correct.

5.    Line 327: “Growing scientific evidence has shown…”. I find the word “demonstrated” is better than “showed” here.